# Training-free Detection of AI-generated images via Cropping Robustness

**Sungik Choi**[1]  **Hankook Lee**[2]  **Moontae Lee**[1,3]
[1]LG AI Research  [2]SungkyunKwan University  [3]University of Illinois Chicago
sungik.choi@lgresearch.ai

## Abstract

AI-generated image detection has become crucial with the rapid advancement of vision-generative models. Instead of training detectors tailored to specific datasets, we study a training-free approach leveraging self-supervised models without requiring prior data knowledge. These models, pre-trained with augmentations like `RandomResizedCrop`, learn to produce consistent representations across varying resolutions. Motivated by this, we propose **WaRPAD**, a training-free AI-generated image detection algorithm based on self-supervised models. Since neighborhood pixel differences in images are highly sensitive to resizing operations, WaRPAD first defines a base score function that quantifies the sensitivity of image embeddings to perturbations along high-frequency directions extracted via Haar wavelet decomposition. To simulate robustness against cropping augmentation, we rescale each image to a multiple of the model's input size, divide it into smaller patches, and compute the base score for each patch. The final detection score is then obtained by averaging the scores across all patches. We validate WaRPAD on real datasets of diverse resolutions and domains, and images generated by 23 different generative models. Our method consistently achieves competitive performance and demonstrates strong robustness to test-time corruptions. Furthermore, as invariance to `RandomResizedCrop` is a common training scheme across self-supervised models, we show that WaRPAD is applicable across self-supervised models.

## 1 Introduction

AI-generated image detection aims to design reliable metrics that can distinguish between real and synthetically generated images. This task has become increasingly critical with the advent of highly capable text-to-image (T2I) generative models that can produce photorealistic outputs, which may be exploited for vicious purposes (*e.g.*, fake news [1], deepfakes [2]). Most existing detection approaches [3, 4] are trained to recognize specific real data distributions (*e.g.*, ImageNet [5], LSUN [6]). However, the scope of real image distributions that current detection approaches can effectively cover remains extremely limited compared to the diversity of generated images. Furthermore, a training dataset for detection may contain artifacts (*e.g.*, WebP compression in LSUN), which may lead the detector to overfit the artifacts and may fail to be robust in test-time corruptions [7]. Consequently, there is a growing need for detection methods that can operate universally across diverse domains without relying on the constraints of predefined real image distributions.

In response to this growing demand, this paper focuses on *training-free* detection methods where no prior knowledge of real image distributions is given. Prior training-free detection methods have design score functions based on representations extracted from large-scale pre-trained foundation models. One representative line of work leverages the representation of latent diffusion models (LDMs) (*e.g.*, Stable Diffusion [8], Midjourney [9]). For instance, AEROBLADE [10] detects LDM-generated images by measuring the autoencoder reconstruction loss, while Manifold Bias [11] proposes a

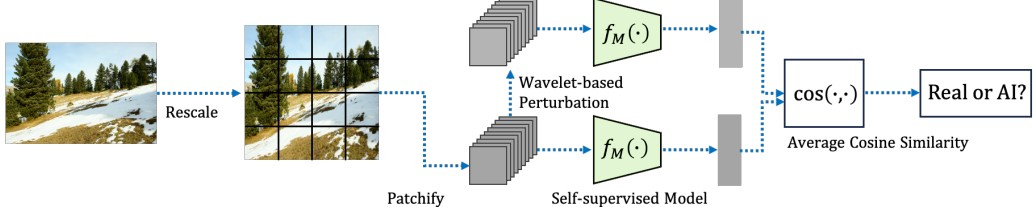

Figure 1: **Conceptual illustration of our method WaRPAD.** We first rescale and patchify the given image to the batch of patches. Then, we perturb the patches on the high-frequency direction of Haar wavelet decomposition. Our final score function is the averaged cosine similarity between the perturbed and non-perturbed patches' features through the self-supervised model.

curvature-based metric in the latent space of the LDM. However, the performance of these approaches is often tied closely to the choices of the LDMs, and their generalizability to other generative models remains uncertain.

Another stream of training-free research utilizes representations from self-supervised models, such as DINOv2 [12]. These models benefit from pre-training on a wide range of real-world images and are often applied as generalist models for various downstream tasks [13, 14]. Some prior works [15, 16] attempt to utilize self-supervised models on AI-generated image detection by introducing simple perturbations (*e.g.*, Gaussian noise or Gaussian blurring) and measure the cosine similarity between the original and perturbed image embeddings. However, their performance often lags behind the diffusion-based training-free baselines (see Section 4.2).

**Contribution.** In this study, we approach AI-generated image detection from a data augmentation perspective, utilizing foundation models that have been pre-trained on real images. Our primary motivation stems from the observation that these models are trained to produce consistent embedding representations between an original image and its randomly cropped and resized variants. We hypothesize that embeddings of AI-generated images exhibit lower robustness to such `RandomResizedCrop` (RRC) transformations than those of real images.

To investigate this hypothesis, we examine how RRC affects the high-frequency components of images, as obtained through wavelet decomposition. Notably, we observe that even when the cropped image closely matches the original in size, RRC introduces substantial variations in the difference in the neighborhood pixels. This indicates that RRC acts as an effective perturbation on the high-frequency components extracted via Haar wavelet decomposition. Given that foundation models are typically trained to be invariant to such perturbations on the real images, we propose a detection score function that quantifies embedding sensitivity to high-frequency distortions as a signal for distinguishing real and synthetic images.

Furthermore, to simulate the effects of RRC in a more structured and deterministic manner, we resize each image to a multiple ($\times K^2$) of the default input resolution and partition it into $K^2$ patches of the default resolution. We then apply the proposed score function to each patch and aggregate the results. This patch-based perturbation consistently reveals a greater discrepancy in AI-generated images, demonstrating the effectiveness of our method. Figure 1 shows the computation of our unified method, **WaRPAD**: **Wa**velet, **R**esizing, and **P**atchifying for **A**I-generated image **D**etection.

We conduct extensive evaluations of WaRPAD across multiple AI-generated image detection benchmarks. These benchmarks span various generative model types, including LDMs, proprietary models (*e.g.*, Firefly [17], Dall-E [18]), and generative adversarial network (GAN) [19] architectures, as well as multiple image domains. Our method consistently outperforms other training-free baselines in all settings. Notably, we observe an improvement of **6.5 $\sim$ 24.7% in AUROC** over prior methods based on the same DINOv2 model. Furthermore, we evaluate the robustness of our method against various image corruptions and show that it maintains competitive performance under such conditions, surpassing other detection methods.

In brief, our contributions are summarized as follows.

- We propose WaRPAD that applies a self-supervised foundation model's robustness on RRC augmentation for detecting AI-generated images (Section 3).

- WaRPAD outperforms existing training-free AI-generated image detection methods in every benchmark consistently (Section 4.2). Furthermore, WaRPAD is robust to test-time corruption of the examined images (Section 4.3).
- Our analysis suggests that a self-supervised model trained to be invariant under RRC can be applied for AI-generated image detection, supporting the generalizability of WaRPAD (Section 4.3).

## 2 Preliminary

First, we introduce the AI-generated image detection framework. Furthermore, we discuss the self-supervised models and their key data augmentation strategy. For the broader overview, we refer to Deng et al. [20] and Uelwer et al. [21] for the comprehensive survey.

### 2.1 AI-generated Image Detection

AI-generated image detection aims to design a scoring function $S(\mathbf{x})$ that determines whether a given image $\mathbf{x}$ has been acquired from the real world (*i.e.*, $S(\mathbf{x}) \geq \tau$) or generated by a generative model (*i.e.*, $S(\mathbf{x}) < \tau$). While most existing approaches train $S(\mathbf{x})$ using labeled datasets containing both real and synthetic images, we assume a more practical setting in which such training data is not available. This setup allows us to assess the generalizability of detection methods to previously unseen data distributions.

Recently, this generalizability to unseen distributions has been discussed in training-based approaches as well. ZED [3] proposes an entropy-based score that can be trained solely on real image distributions without requiring any synthetic examples. Cozzolino et al. [22] train a Linear SVM classifier on CLIP [23] embeddings extracted from image pairs (*e.g.*, MS-COCO [24] and LDM-generated [8]) and evaluate its performance on unseen images (*e.g.*, Raise-1K [25] and Firefly [17]). Rajan and Lee [7] improve robustness to post-processed images by retraining the final layer of the pre-trained detection model. Extending beyond these approaches, we focus on a training-free detection setting in which no real or fake data is provided during the design of the detection score.

Training-free detection aims to construct a universal score based on the outputs of a pre-trained foundation model. Such a foundation model may either be a generative model itself (*e.g.*, LDM [8]) or a model trained on diverse real-world images (*e.g.*, DINOv2 [12], CLIP [23]). In this work, we adopt the latter approach, leveraging models pre-trained on real data to guide our detection framework.

### 2.2 Self-supervised Models

Self-supervised models aim to learn robust representations from large-scale unannotated images that can be generalized to a wide range of downstream tasks. Self-supervised learning methods apply various augmentations, often referred to as views, to a single image and train the model to maximize the similarity between outputs corresponding to different views. This paradigm has been implemented in various forms, including contrastive learning (*e.g.*, SimCLR [26], Moco [27]), clustering-based (*e.g.*, SwaV [28]), and knowledge distillation (*e.g.*, DINO [29]). Since providing a comprehensive overview of self-supervised learning (SSL) is beyond the scope of this paper, we focus on the DINOv2 [12] model and the augmentation strategies.

DINOv2 is a Vision Transformer [30] model trained on a web-scale LVD-142M dataset, building upon the iBOT [31] framework. DINOv2 tokenizes each input image and outputs patch-level embeddings along a `[CLS]` token embedding that summarizes the entire image. The model is trained using a teacher-student framework, where the student network is optimized to maximize the similarity between its output and the output of the teacher network given relatively mild augmentations. The teacher network is updated via an exponential moving average of the student parameters.

A core component of the data augmentation applied to both networks is `RandomResizedCrop` (RRC), which randomly crops the region from the input image and resizes it to the given resolution. This operation enforces spatial invariance by encouraging the model to recognize that different subregions of an image correspond to the same underlying semantic content. Due to its effectiveness, RRC has become a standard augmentation technique across many SSL frameworks. In this work, we adopt RRC as the key motivation for designing our AI-generated image detection method.

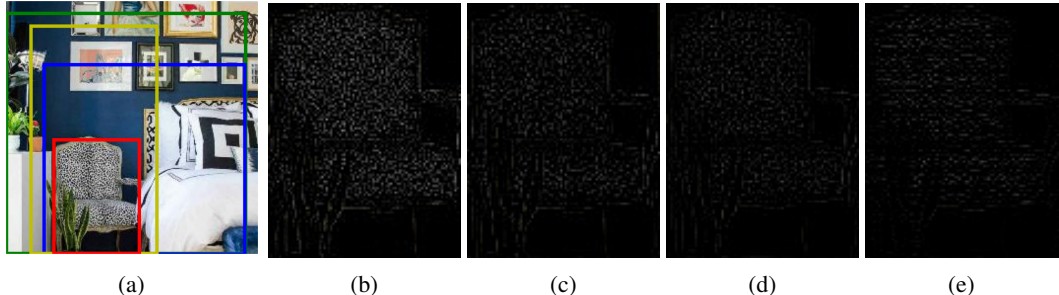

|  (a)  |  (b)  |  (c)  |  (d)  |  (e)  |

Figure 2: **Motivation for the Haar-wavelet perturbation sensitivity score. (a)**: The original image along with the designated region for high-frequency visualization, marked in **red**. To simulate the effect of `RandomResizedCrop` (RRC), we apply cropping regions indicated in green, blue, and yellow. **(b)**: The high-frequency component of the original (uncropped) image obtained via Haar wavelet decomposition. **(c)**, **(d)**, **(e)**: The corresponding high-frequency components of the RRC-transformed images, where the cropping regions are defined by the green, blue, and yellow boxes, respectively.

## 3 Method

This section introduces our method, WaRPAD, inspired by the `RandomResizedCrop` (RRC) operation. We first propose a score function that measures the sensitivity of self-supervised model features to perturbations along high-frequency directions induced by wavelet decomposition (Section 3.1). Subsequently, we simulate local robustness by introducing a rescaling-and-patching paradigm that partitions resized images into subregions for evaluation (Section 3.2).

### 3.1 Base Detection Score

We hypothesize that measuring an image's robustness to RRC can serve as an effective score for detecting AI-generated images. However, RRC involves random cropping over a broad hyperparameter space, which introduces variance and may discard key image content of the original image. To address this, we do not apply RRC directly but instead propose a score function that approximates its effect. Specifically, we examine how RRC alters the high-frequency components of the image using wavelet decomposition, motivated by the previous works that AI-generated images exhibit different high-frequency information [32, 33].

For the analysis, we visualize the high-frequency components obtained via Haar wavelet decomposition under multiple instances of RRC, as illustrated in Figure 2a. Figure 2b presents the high-frequency component within the red-marked region of the original image, whereas Figures 2c, 2d, and 2e show the corresponding components after applying RRC with green, blue, and yellow cropping regions, respectively. These comparisons reveal that RRC introduces substantial variations in the high-frequency components, even when the cropped region closely matches the original image in size. Based on this observation, we define our base score function as the model's sensitivity to perturbations along the high-frequency directions, formally expressed as follows:

$$\text{HFwav}(\mathbf{x}) = \frac{\mathbf{f_M}(\mathbf{x}) \cdot \mathbf{f_M}(\mathbf{x} - \alpha \text{HF}(\mathbf{x}))}{\|\mathbf{f_M}(\mathbf{x})\| \|\mathbf{f_M}(\mathbf{x} - \alpha \text{HF}(\mathbf{x}))\|}, \tag{1}$$

where $\mathbf{f_M}$ denotes the feature output of the self-supervised model M (e.g., the `[CLS]` token output of DINOv2), HF represents the high-frequency component derived from wavelet decomposition, and $0 < \alpha < 1$ is the perturbation weight. The sensitivity score function HFwav quantifies the change of model M to perturbations along high-frequency directions. Our central hypothesis is that model M, having been trained on real images, is encouraged to be invariant to such high-frequency perturbations. Accordingly, we expect the HFwav score to be higher for real images than the AI-generated images.

### 3.2 WaRPAD

We further propose a test-time augmentation strategy inspired by RRC. Our main idea is to deterministically simulate multiple instances of RRC by explicitly rescaling the image and thereby patchifying

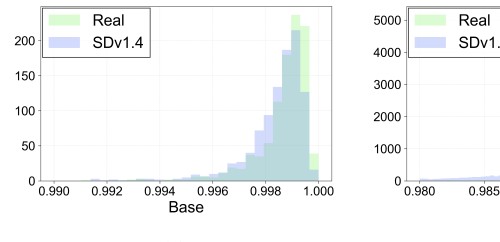 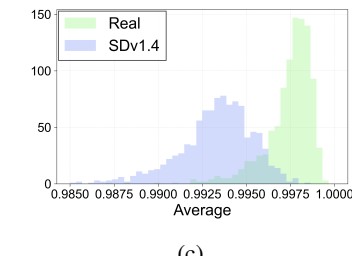

|         (a)         |         (b)         |         (c)         |

Figure 3: **Effect of** `RescaleNPatchify`. **(a)**: Histogram of real and SDv1.4-generated data examined by HFwav. **(b)**: Histogram of real and SDv1.4-generated data examined on patches augmented by `RescaleNPatchify`. **(c)**: Histogram of real and SDv1.4-generated data examined on our WaRPAD score function.

Table 1: **Benchmarks in the main experiment.**

| Benchmark | Real Dataset | # of Generative Models | Input Resolution | # of Test Images per Dataset |
|---|---|---|---|---|
| Synthbuster [34] | Raise-1K [25] | 9 (LDM, Proprietary Model) | Varying ($256 \times 256 \sim 4928 \times 3264$) | 1000 |
| GenImage [35] | ImageNet [5] | 8 (GAN, Diffusion Model, LDM) | Varying ($128 \times 128 \sim 1024 \times 1024$) | $6000 \sim 8000$ |
| Deepfake-LSUN-Bedroom [36] | LSUN [6] | 10 (GAN, Diffusion Model) | $256 \times 256$ | 10000 |

the image to patches with the same sizes as follows:

$$\texttt{RescaleNPatchify}(\mathbf{x}) = \texttt{Patchify}\left(\texttt{Rescale}(\mathbf{x}, d_{\text{rescale}}), d_{\text{patch}}\right), \qquad (2)$$

where $d_{\text{rescale}}$ and $d_{\text{patch}}$ are the dimension of the rescaled image and the dimension of the patch, $\texttt{Rescale}(\mathbf{x}, d)$ is the image rescaling operation to dimension $d \times d$, and $\texttt{Patchify}(\mathbf{x}, d)$ is the image patchifying operation to patch dimension $d \times d$, respectively.

The `RescaleNPatchify` operation results in $n_{\text{patch}} = (\frac{d_{\text{rescale}}}{d_{\text{patch}}})^2$ number of patches with $d_{\text{patch}} \times d_{\text{patch}}$ dimension. We now examine the proposed sensitivity score in Section 3.1 and average the sensitivity score to output the unified score function as follows:

$$\text{WaRPAD}(\mathbf{x}) = \frac{1}{n_{\text{patch}}} \sum_{\mathbf{x}_{\text{patch}} \in \texttt{RescaleNPatchify}(\mathbf{x})} \text{HFwav}(\mathbf{x}_{\text{patch}}) \qquad (3)$$

Our proposed WaRPAD examines the sensitivity score in the resized subregion of the images, which are multiple instances of RRC with the area of $\frac{1}{n_{\text{patch}}}$ resized to a dimension of $d_{\text{patch}} \times d_{\text{patch}}$. Since the model M is trained to be invariant under various RRC instances that include crops similar to the patch, we expect the base score HFwav evaluated on the patch to be still robust. On the other hand, We hypothesize the model output of the AI-generated image will deviate in the patches, hence our WaRPAD further improves on detecting AI-generated images. We verify our hypothesis by examining the WaRPAD and our base score in 1000 real RAISE-1k [25] data and 1000 SDv1.4-generated data in the Synthbuster benchmark [34]. We select $d_{\text{rescale}} = 1344$ and $d_{\text{patch}} = 224$.

We show the histogram of real and AI-generated images under our base score on the image, base score on the patch, and the averaged WaRPAD score in Figure 3a, 3b, and 3c, respectively. AI-generated images lose their robustness in our wavelet-based high-frequency perturbation when examined in patches generally. The discrimination between AI-generated images and real images is strengthened in our final score induced by averaging the score function across patches.

## 4 Experiment

We now evaluate WaRPAD's efficacy on AI-generated image detection. We first introduce the benchmarks and generative models for each benchmark as well as the baseline methods (Section 4.1). We report the performance of WaRPAD across these benchmarks (Section 4.2). Finally, we present detailed ablation studies of WaRPAD and test its robustness in corruptions (Section 4.3).

### 4.1 Experiment Settings

**Datasets.** We first test WaRPAD in Synthbuster [34] benchmark based on RAISE-1k dataset [25] where 9 generative models are applied: Firefly [17], GLIDE [38], SDXL [39], SDv2, SDv1.3, SDv1.4

Table 2: **AI-generated image detection performance (AUROC) of WaRPAD and baselines in the Synthbuster [34] benchmark. Bold** and underline denotes the best method and the second best methods, respectively.

| Method | Firefly | GLIDE | SDXL | SDv2 | SDv1.3 | SDv1.4 | DALL-E 3 | DALL-E 2 | Midjourney | Mean |
|---|---|---|---|---|---|---|---|---|---|---|
| Training-based Methods | | | | | | | | | | |
| AIDE [4] | 0.165 | 0.780 | 0.835 | 0.642 | 0.946 | 0.933 | 0.415 | 0.426 | 0.688 | 0.648 |
| FatFormer [37] | 0.586 | 0.718 | 0.707 | 0.513 | 0.486 | 0.500 | 0.186 | 0.571 | 0.374 | 0.516 |
| Training-free Methods | | | | | | | | | | |
| RIGID [15] | 0.519 | 0.868 | 0.757 | 0.615 | 0.448 | 0.446 | 0.442 | 0.596 | 0.593 | 0.587 |
| MINDER [16] | 0.440 | 0.568 | 0.472 | 0.721 | 0.656 | 0.668 | 0.346 | 0.445 | 0.345 | 0.518 |
| AEROBLADE [10] | 0.592 | 0.954 | 0.668 | 0.567 | 0.950 | 0.950 | **0.486** | 0.392 | **0.769** | 0.703 |
| Manifold Bias [11] | 0.493 | 0.779 | 0.562 | 0.749 | 0.544 | 0.549 | 0.379 | 0.607 | 0.424 | 0.565 |
| WaRPAD (ours) | **0.927** | **0.999** | **0.830** | **0.775** | **0.959** | **0.958** | 0.422 | **0.930** | 0.702 | **0.834** |

Table 3: **AI-generated image detection performance (AUROC) of WaRPAD and baselines in the GenImage [35] benchmark. Bold** and underline denotes the best and second best methods, respectively.

| Method | ADM | BigGAN | GLIDE | Midjourney | SDv1.4 | SDv1.5 | VQDM | Wukong | Mean |
|---|---|---|---|---|---|---|---|---|---|
| Training-based Methods | | | | | | | | | |
| AIDE [4] | 0.921 | 0.920 | 0.987 | 0.959 | 1.000 | 1.000 | 0.965 | 1.000 | 0.969 |
| FatFormer [37] | 0.903 | 0.995 | 0.951 | 0.579 | 0.780 | 0.776 | 0.967 | 0.824 | 0.847 |
| Training-free Methods | | | | | | | | | |
| RIGID [15] | 0.874 | 0.974 | 0.952 | 0.778 | 0.682 | 0.682 | 0.915 | 0.699 | 0.820 |
| MINDER [16] | 0.768 | 0.681 | 0.582 | 0.450 | 0.607 | 0.596 | 0.882 | 0.676 | 0.655 |
| AEROBLADE [10] | 0.856 | 0.981 | 0.989 | 0.918 | 0.982 | 0.984 | 0.732 | **0.983** | 0.928 |
| Manifold Bias [11] | 0.727 | 0.925 | 0.852 | 0.510 | 0.675 | 0.673 | 0.874 | 0.653 | 0.736 |
| WaRPAD (ours) | **0.986** | **0.998** | **0.991** | 0.810 | 0.940 | 0.936 | **0.981** | 0.924 | **0.946** |

Table 4: **AI-generated image detection performance (AUROC) of WaRPAD and baselines in the Deepfake-LSUN-Bedroom [36] benchmark. Bold** and underline denotes the best method and the second best methods, respectively.

| Method | ADM | DDPM | Diff-ProjectedGAN | Diff-StyleGAN2 | IDDPM | LDM | PNDM | ProGAN | ProjectedGAN | StyleGAN | Mean |
|---|---|---|---|---|---|---|---|---|---|---|---|
| Training-based Methods | | | | | | | | | | | |
| AIDE [4] | 0.636 | 0.722 | 0.860 | 0.951 | 0.679 | 0.807 | 0.941 | 0.899 | 0.910 | 0.840 | 0.825 |
| FatFormer [37] | 0.745 | 0.709 | 0.998 | 1.000 | 0.824 | 0.944 | 0.999 | 1.000 | 0.999 | 0.988 | 0.921 |
| Training-free Methods | | | | | | | | | | | |
| RIGID [15] | 0.742 | 0.887 | 0.937 | 0.914 | 0.855 | 0.846 | 0.843 | 0.957 | 0.944 | 0.681 | 0.861 |
| MINDER [16] | 0.706 | 0.796 | 0.973 | 0.942 | 0.782 | 0.844 | 0.896 | 0.970 | 0.973 | 0.805 | 0.869 |
| AEROBLADE [10] | 0.545 | 0.741 | 0.488 | 0.534 | 0.656 | 0.595 | 0.382 | 0.454 | 0.490 | 0.342 | 0.522 |
| Manifold Bias [11] | **0.788** | 0.905 | 0.968 | 0.943 | 0.888 | 0.928 | 0.891 | **0.996** | 0.978 | **0.912** | 0.920 |
| WaRPAD (ours) | 0.785 | **0.937** | **0.988** | **0.965** | **0.908** | **0.940** | **0.970** | 0.995 | **0.986** | 0.870 | **0.934** |

[8], DALL-E 3 [40], DALL-E 2 [18], and Midjourney [9]. We also test WaRPAD in GenImage [35] benchmark where the real data is from the ImageNet [5] dataset and 8 generative models are applied for fake image generation: ADM [41], BigGAN [42], GLIDE, Midjourney, SDv1.4, SDv1.5, VQDM [43], and Wukong [44]. Finally, we test on the deepfake-LSUN-bedroom benchmark [36] containing 10 generative models: ADM, DDPM [45], Diff-ProjectedGAN, Diff-StyleGAN2 [46], IDDPM [47], LDM [8], PNDM [48], ProGAN [49], ProjectedGAN [50], and StyleGAN [51]. We summarize the main information of these benchmarks in Table 1.

**Baselines.** We consistently compare against available training-free baselines. AEROBLADE [10] and Manifold Induced Bias [11] apply SD for the examination. Consistent with the proposal of Brokman et al. [11], we apply SDv1.4 for the inspection model. We also compare against RIGID [15] and MINDER [16], which apply DINOv2 for AI-generated image detection. We follow the original hyperparameter settings from the paper. To further examine the efficacy of training-free setting, we also compare with the leading training-based methods [4, 37] for comparison. Specifically, we examine the pre-trained checkpoint of each method in each benchmark without further training. Note that AIDE is trained on real ImageNet data and SDv1.4-generated data, while FatFormer is trained on LSUN data and ProGAN-generated data.

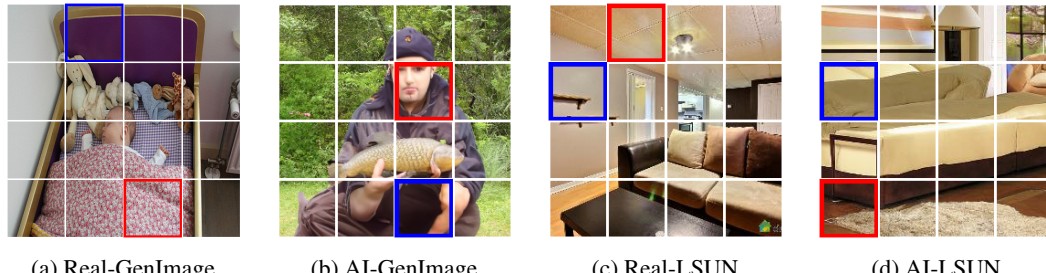

| (a) Real-GenImage | (b) AI-GenImage | (c) Real-LSUN | (d) AI-LSUN |

Figure 4: **Visualization of the HFwav score across patches.** We show the patch with the highest score in red and lowest score in blue. Each image is from ImageNet **(a)**, ADM-generated GenImage **(b)**, LSUN **(c)**, and ADM-generated Deepfake-LSUN-Bedroom dataset **(d)**, respectively.

**Implementation Details.** The performance of all methods is reported by the area under the ROC curve (AUROC). Consistent with RIGID and MINDER, we use the DINO-ViT-L14 model as the base model. We use the Haar wavelet with a 2-level decomposition to extract the high-frequency information. We also set $d_{patch}$ and $\alpha$ to 224 and 0.1 throughout all experiments. For the rescaling dimension $d_{rescale}$, we set 896 for the GenImage and Deepfake-LSUN-bedroom benchmark and 1344 for the Synthbuster benchmark. We implement our code in the Pytorch [52] framework. All experiments are done on a single A100 GPU.

## 4.2 Main Results

Tables 2, 3, and 4 report the performance of WaRPAD compared to other training-free approaches in Synthbuster, GenImage, and Deepfake-LSUN-bedroom benchmark, respectively. WaRPAD achieves the best performance on all benchmarks on average. Notably, WaRPAD consistently outperforms RIGID and MINDER by a wide margin of over 6% in AUROC.

On the other hand, diffusion-model-based methods fail to show consistent performance compared to WaRPAD. For example, while AEROBLADE is competitive to WaRPAD in the GenImage benchmark, where some generative models share the same autoencoder with the inspected SDv1.4 model (*e.g.*, SDv1.5, Wukong), its performance deteriorates on detecting proprietary models (*e.g.*, Firefly, DALL-E 2) or GAN-based models. On the other hand, Brokman et al. [11] can efficiently perform in the Deepfake-LSUN-Bedroom benchmark but fails on the GenImage and Synthbusters benchmarks.

Finally, training-based methods fail to generalize to unobserved real data distribution and underperform over our WaRPAD. Specifically, while AIDE performs well on detecting SD-generated data in the GenImage benchmark, it underperforms on other generative models and Deepfake-LSUN-Bedroom/ Synthbuster benchmarks. A similar phenomenon occurs in FatFormer, which shows underwhelming performance in GenImage/Synthbuster benchmarks. On the other hand, WaRPAD shows the best performance in general, even improving over the FatFormer in LSUN-based benchmark.

We also visualize the patch-wise score information of the real and AI-generated images. Figure 4 shows the patch with the highest HFwav score (denoted as red) and the patch with the lowest score (denoted as blue) in the real and AI-generated data in GenImage and Deepfake-LSUN-Bedroom benchmark, respectively. While HFwav assigns high scores on patches with rich texture information, the region does not always align with the semantics of the image (*e.g.*, Figure 4c).

## 4.3 Analysis

We formulate our analysis to validate the following questions:

- Does each component of WaRPAD contribute to consistent performance gains?
- Is WaRPAD robust to design choices, hyperparameter ablation, and test-time perturbations?
- Is WaRPAD effective for other domains?

Table 5: **Ablation Study on each component of WaRPAD.** We report AUROC. Gains are computed against RIGID [15].

| Method | Synthbuster | GenImage | Deepfake-LSUN-Bedroom |
|---|---|---|---|
| RIGID [15] | 0.587 | 0.820 | 0.861 |
| RIGID $\times n_{\text{patch}}$ | 0.589 (+0.2%) | 0.823 (+0.3%) | 0.872 (+1.1%) |
| HFwav | 0.636 (+4.9%) | 0.809 (-1.1%) | 0.890 (+2.9%) |
| RIGID + RescaleNPatchify | 0.656 (+6.9%) | 0.800 (-2.0%) | 0.861 (+0.0%) |
| WaRPAD | 0.834 (+24.7%) | 0.946 (+12.6%) | 0.934 (+7.3%) |

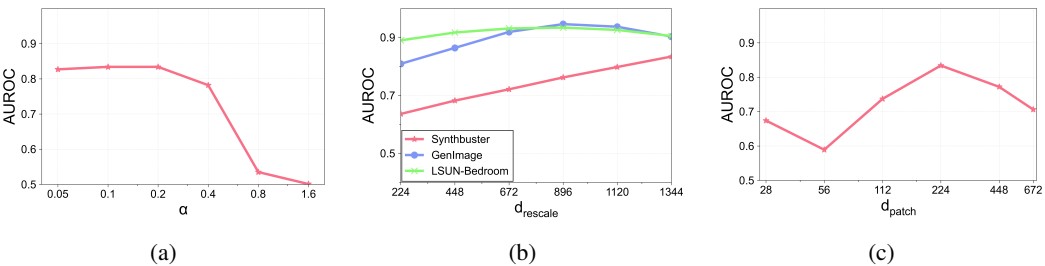

(a)    (b)    (c)

Figure 5: **Hyperparameter analysis of WaRPAD (a)**: AUROC performance of WaRPAD with respect to $\alpha$ in the Synthbuster benchmark. **(b)**: AUROC result with respect to the $d_{\text{rescale}}$. **(c)**: AUROC result with respect to the $d_{\text{patch}}$ in the Synthbuster benchmark.

**Effect of each Component.** We first analyze the effect of our proposed HFwav and RescaleNPatchify operation. We also investigate the synergy of HFwav and RescaleNPatchify by experimenting with RIGID [15] with the same RescaleNPatchify procedure. Further, we experiment with the $n_{\text{patch}}$-ensemble version of RIGID that takes the same computational cost as our WaRPAD. We show the results in Table 5 with a comparison against RIGID [15]. Our perturbation-based metric improves over RIGID in two out of 3 benchmarks. Moreover, RIGID combined with our proposed RescaleNPatchify shows relatively little improvement from RIGID compared to WaRPAD, highlighting the efficacy of both components.

**Hyperparameter Analysis.** We analyze the effect of the hyperparameters of the WaRPAD independently. Figure 5a analyzes the effect of perturbation weight $\alpha$ in the Synthbuster benchmark. We also analyze the effect of the rescaling dimension $d_{\text{rescale}}$ on separate benchmarks in Figure 5b. All hyperparameter choices consistently improve over the base dimension, 224. Finally, as DINOv2 can accept arbitrary patch sizes, we also test WaRPAD in the different patch dimensions $d_{\text{patch}}$. We show the result in Figure 5c where the base choice of 224 performs the best.

**Design Choices.** We first show the AUROC result of WaRPAD across different aggregation rules of the computed patch-wise metric and the backbone DINOv2 version. For the aggregation rule, we test the mean, median, minimum, and maximum across patches. For the backbone model, we experiment with 'ViT-S14', 'ViT-B14', 'ViT-L14', and 'ViT-g14'. We also experiment with the 'ViT-L14' and 'ViT-g14' with register tokens [14].

We show the result in Table 6. Note that Mean and 'ViT-L14' correspond to the results in Section 4.2. In the perspective of the aggregation rule, the mean or median aggregation rule achieves the best performance consistently. On the other hand, concerning the DINOv2 backbone, a larger model size shows better results with the slight exception of the 'ViT-L14' backbone in the GenImage benchmark.

We further experiment with the different choices of the wavelet and the decomposition level of the wavelet decomposition. Apart from the Haar wavelet, we experiment with Daubeches wavelets (db2, db3, db4), Biorthogonal wavelets (bior1.3, bior1.5, bior2.2, bior2.4, bior3.1), and Coiflet wavelets (coif1, coif2, coif3) with decomposition levels from 1 to 3.

We present the results in Table 7, grouping wavelets by the number of vanishing moments in the (synthesis) wavelet function $\psi$. While the Haar wavelet achieves the highest performance, our results indicate that other wavelets with one vanishing moment also perform competitively. In contrast, wavelets with higher vanishing moments tend to induce more structured perturbations on the real images, and we find that DINOv2 is no longer robust to the perturbation. We further report this

Table 6: **Ablation Study on the DINOv2 backbone and patch-wise aggregation rule on WaRPAD.** We report AUROC. **Bold** denotes the best choice.

| Agg Rule | ViT-S14 | ViT-B14 | ViT-L14 | ViT-L14-reg | ViT-g14 | ViT-g14-reg |
|----------|---------|---------|---------|-------------|---------|-------------|
| Mean | 0.76/0.86/0.83 | 0.82/0.92/0.90 | 0.83/**0.95**/0.93 | 0.84/0.94/0.94 | 0.86/0.94/**0.95** | **0.87**/0.94/**0.95** |
| Median | 0.77/0.85/0.81 | 0.83/0.91/0.89 | 0.84/0.94/0.92 | 0.85/0.94/0.93 | 0.85/0.93/**0.95** | **0.87**/0.94/**0.95** |
| Min | 0.71/0.85/0.79 | 0.75/0.89/0.84 | 0.76/0.91/0.88 | 0.77/0.91/0.89 | 0.80/0.90/0.91 | 0.80/0.90/0.90 |
| Max | 0.68/0.74/0.63 | 0.77/0.81/0.77 | 0.78/0.90/0.79 | 0.78/0.90/0.76 | 0.79/0.90/0.87 | 0.80/0.91/0.87 |

Table 7: **AI-generated image detection performance (AUROC) to wavelet choice and decomposition level in the Synthbuster benchmark. Bold** and underline denotes the best and the second best choice. We group the wavelets by the number of vanishing moments on a wavelet function $\psi$.

| | 1 vanishing moment | | | >1 vanishing moments | | | | | | | | |
|-------|------|--------|--------|------|------|------|----------|----------|----------|-------|-------|-------|
| Level | Haar | bior 1.3 | bior 1.5 | db2 | db3 | db4 | bior 2.2 | bior 2.4 | bior 3.1 | coif1 | coif2 | coif3 |
| 1 | 0.745 | 0.715 | 0.701 | 0.458 | 0.505 | 0.527 | 0.474 | 0.483 | 0.486 | 0.480 | 0.481 | 0.487 |
| 2 | **0.834** | 0.591 | 0.827 | 0.512 | 0.491 | 0.508 | 0.512 | 0.487 | 0.472 | 0.533 | 0.506 | 0.501 |
| 3 | 0.787 | 0.585 | 0.569 | 0.466 | 0.490 | 0.460 | 0.476 | 0.490 | 0.486 | 0.498 | 0.457 | 0.467 |

Table 8: **AI-generated image detection performance (AUROC) in the Synthbuster benchmark under different backbones.**

| | Invariant to RRC | | | | Others | |
|--------|------------|----------|----------|-----------|----------|------------|
| Method | DINOv2 [12] | CLIP [23] | SwaV [28] | DINO [29] | BeiT [53] | ViTMAE [54] |
| RIGID | 0.587 | 0.561 | 0.542 | 0.480 | 0.619 | 0.531 |
| MINDER | 0.518 | 0.583 | 0.542 | 0.478 | 0.473 | 0.545 |
| WaRPAD | 0.834 | 0.802 | 0.743 | 0.707 | 0.486 | 0.620 |

phenomenon in the Appendix, where we include histograms of the score distributions for other wavelets for both real and AI-generated images.

**Robustness to Corruptions.** Our WaRPAD is based on the self-supervised vision model, which may learn robust representations due to training on a wide range of perturbations (*e.g.*, `ColorJitter`, `RandomSolarize`). Hence, we test WaRPAD's robustness on the corruption of the input images, both real and AI-generated. For comparison, we also test RIGID, MINDER, and AEROBLADE on the same corruption. We report the average performance on the GenImage benchmark with JPEG compression, center crop and resizing, and Gaussian noise corruptions with varying degrees.

Figure 6 shows the performance of WaRPAD as well as the training-free baselines under corruption in the GenImage benchmark. WaRPAD achieves competitive performance over the baseline in every corruption consistently. On the other hand, AEROBLADE's performance quickly degrades compared to WaRPAD in high-level corruption. Experiments in other benchmarks in the Appendix exhibit consistent behaviors.

**Extension to other backbones.** The result in DINOv2 shows that our proposed metric can perform not only in in-domain datasets (*e.g.*, ImageNet) but also in datasets unobserved in the training phase (*e.g.*, RAISE-1k, LSUN-Bedroom). We further experiment WaRPAD with other backbones of self-supervised models trained to be invariant under RRC. For the model, we select CLIP [23], SwaV [28], and DINO [29]. We also test other models that use RRC only as data augmentation: BeiT [53] and ViTMAE [54]. We also specify the details of the tested models in the Appendix. We further include MINDER and RIGID for each backbone as a comparison. We do not change any hyperparameters.

As shown in Table 8, our WaRPAD consistently outperforms RIGID and MINDER when the backbone model is trained to be invariant under RRC. However, the gain of WaRPAD is less prominent when tested in the masked-image-modeling-based backbones and even underperforms over RIGID when the backbone model is BeiT.

**Examination on other domain data.** While we thoroughly evaluate WaRPAD on various benchmarks, such benchmarks are from the natural image domain. Motivated by the recent practice [55] that evaluates AI-generated image detectors outside the natural image domain, we further test the

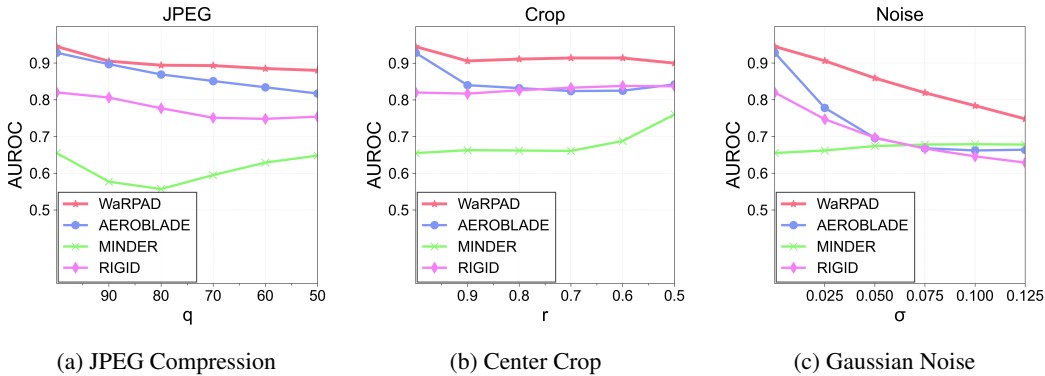

(a) JPEG Compression        (b) Center Crop        (c) Gaussian Noise

Figure 6: **Robustness of WaRPAD in corruptions.** We test the AUROC performance of WaRPAD, AEROBLADE, MINDER, and RIGID in corrupted test images in the GenImage benchmark.

Table 9: **Zero-shot performance of AI-generated image detection methods on the Art domain.** We report AUROC.

| Method | FatFormer [37] | RIGID [15] | MINDER [16] | WaRPAD (ours) |
|--------|----------------|------------|-------------|---------------|
| AUROC  | 0.531          | 0.725      | 0.365       | 0.765         |

performance of WaRPAD on the Art domain. Since the FakeART benchmark in [55] is not public except for the information that they use the WikiArt[1] dataset for the real data, we instead download data from the Kaggle webpage[2] with 10821/10821 real and AI-generated data available, where the real data is from the WikiArt dataset. For comparison, we evaluate our method against FatFormer, RIGID, and MINDER on the Art domain, and report the results in Table 9. This demonstrates the effectiveness of WaRPAD under distribution shift.

## 5   Conclusion

We propose WaRPAD, an effective training-free AI-generated image detection method motivated by `RandomResizedCrop`, the core data augmentation scheme of self-supervised methods. WaRPAD shows improved performance and robustness against existing training-free methods. The main advantage of the WaRPAD is the ubiquity of the `RandomResizedCrop`, which enables WaRPAD applicable to various self-supervised models as the backbone. Since WaRPAD applies to vision-text trained encoders (*e.g.*, CLIP [23]), our method can be extended to detecting multimodal AI-generated data. We leave this direction to future work.

**Broader Impact.** WaRPAD offers effective training-free detection of AI-generated images in the wild. This provides a *tabula rasa* defense against the improper use of generative models, including fraud or manipulation of AI-generated images. Furthermore, our method can be applied to filter out AI-generated images from the web-scale image data.

**Limitations.** Our `RescaleNPatchify` procedure induces extra computation costs due to the number of patches, although these patches can be computed in a batch-wise manner. Furthermore, WaRPAD is dependent on the choice of the backbone self-supervised model, and may not generalize to high-resolution real/AI-generated images outside the scope of the backbone model. Finally, when future generative models can succeed in faithfully generating realistic images (including high-frequency components) enough to fool the pre-trained foundation model, our approach can be less effective. One promising direction is to utilize recent multimodal foundation models, which may show enhanced understanding of AI-generated images. We leave this direction to future work.

---

[1] https://wikiart.org
[2] https://www.kaggle.com/datasets/doctorstrange420/real-and-fake-ai-generated-art-images-dataset

## Acknowledgments and Disclosure of Funding

This work is fully supported by LG AI Research.

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

**Algorithm 1** WaRPAD (PyTorch-like Pseudo-code)

```
# f(x): normalized [cls] token output of self-supervised model
# alpha: weight of perturbation
# DWTForward, DWTInverse: forward and inverse discrete wavelet transform
# Sim: cosine similarity function

def HFwav(x):
    x_low, x_high = DWTForward(x)
    N_perturb = DWTInverse([torch.zeros_like(x_low), x_high])
    feat_original = f(x)
    feat_perturb = f(x - alpha * N_perturb)
    return Sim(feat_original, feat_perturb)

def WaRPAD(x):
    x_patch = RescaleNPatchify(x)
    f_patch = HFwav(x_patch)
    return f_patch.mean()
```

# A    Appendix

## A.1    Pseudocode of WarPAD

We show the Pytorch-like pseudocode of WaRPAD in Algorithm 1. Note that all operations allow batch-wise computation, hence we can process the input patches in a batch-wise manner.

## A.2    Further information of the Experiment settings.

**Synthbuster.** The Synthbuster benchmark consists of 1000 real RAISE-1k images and 9000 AI-generated images consisting of scene and art images under the 'CC BY-NC-SA 4.0' license. We download all real [3] and AI-generated datasets [4] in the URL via the author's official repository.

**GenImage.** The GenImage benchmark consists of ImageNet real data and AI-generated data consisting of 8 different generative models under the 'CC BY-NC-SA 4.0' license. Each test consists of pairs of real and AI-generated image pairs, where the size is 6000+6000 except of SDv1.5, where the size is 8000+8000. We download the datasets via the author's official repository [5].

**Deepfake-LSUN-Bedroom.** The Deepfake-LSUN-Bedroom benchmark consists of 10000 real LSUN-Bedroom images and $10 \times 10000$ AI-generated data where the model is trained to generate LSUN-Bedroom-like images. We download the datasets via the author's official repository [6] under the 'CC BY 4.0' license.

**Baselines.** We follow the author's implementation for the AEROBLADE [7] and Manifold Bias [8], respectively. Since the original implementation of the AEROBLADE operates on the fixed dimension, extension to data with arbitrary size (*e.g.*, Synthbuster, GenImage) is not trivial. Our finding is that the preservation of the original dimension is crucial for the performance, hence we chose to center-crop or resize the image to the fixed dimension of the autoencoder dimension whether the image is larger or smaller than the autoencoder default dimension, respectively. On the other hand, we have not found any official implementation of the authors on the RIGID and MINDER. Instead, we manage to reproduce the RIGID in the consistent setting of our WaRPAD. Note that RIGID and MINDER propose to resize the image to the default resolution of DINOv2, which is $224 \times 224$.

**Experiment Settings.** Most experiments are deterministic since they operate on deterministic operations. A slight exception is RIGID, where the Gaussian noise augmentation is done on the

---

[3]https://loki.disi.unitn.it/RAISE/download.html

[4]https://zenodo.org/records/10066460

[5]https://github.com/GenImage-Dataset/GenImage

[6]https://zenodo.org/records/7528113

[7]https://github.com/jonasricker/aeroblade

[8]https://tinyurl.com/zeroshotimplementation

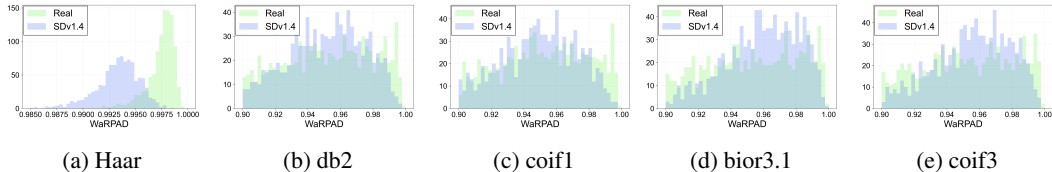

| (a) Haar | (b) db2 | (c) coif1 | (d) bior3.1 | (e) coif3 |

Figure 7: **Histogram of other wavelet.** We show the results on Haar, db2, coif1, bior3.1, and coif3 wavelet, respectively.

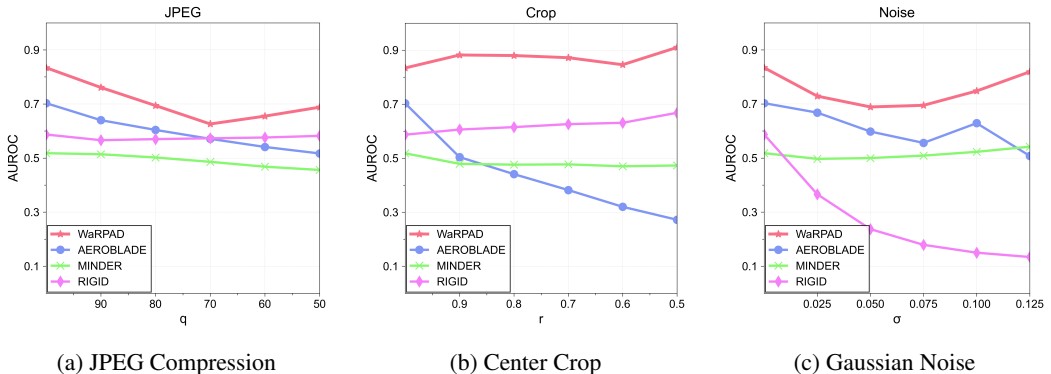

| (a) JPEG Compression | (b) Center Crop | (c) Gaussian Noise |

Figure 8: **Robustness of WaRPAD in corruptions.** We test the AUROC performance of WaRPAD, AEROBLADE, MINDER, and RIGID in corrupted test images in the Synthbuster benchmark.

image. However, our experiments in Table 5 show that RIGID with multiple runs does not change much performance against RIGID with a single seed.

**Additional Backbones.** All pre-trained backbones are accessible and downloadable. For the CLIP model, we use "clip-vit-base-32" [9] for the base model. We use SwaV on the Resnet50 [56] backbone [10] and DINO of "vit-s16" version [11] pre-trained on the ImageNet dataset. We use the "vit-mae-base" model for the ViTMAE backbone [12] and "beit-base-patch16-224" model for the BeiT backbone [13].

### A.3 Histogram of other wavelet choices.

We show the computed histogram of WaRPAD on other wavelets (db2, coif1, bior3.1, and coif3), on real and SDv1.4-generated images computed in the Synthbuster benchmark in Figure 7. Results show DINOv2 model loses its robustness in other wavelet choices, especially wavelets with more vanishing moments.

### A.4 Further robustness experiments

We further include the performance of WaRPAD, AEROBLADE, MINDER, and RIGID in the Synthbuster benchmark in Figure 8 consistent to Figure 6. The trend is consistent, where the WaRPAD performs the best.

---

[9] https://huggingface.co/openai/clip-vit-base-patch32
[10] https://github.com/facebookresearch/swav
[11] https://github.com/facebookresearch/dino
[12] https://huggingface.co/docs/transformers/en/model_doc/vit_mae
[13] https://huggingface.co/docs/transformers/en/model_doc/beit

