# OpenReview forum: "Training-free Detection of AI-generated images via Cropping Robustness"
_NeurIPS.cc/2025/Conference — NeurIPS 2025 poster_

### Official Review · Reviewer_g2xN · 2025-06-28

**Clarity:** 3
**Significance:** 3
**Originality:** 3
**Rating:** 4
**Confidence:** 4

**Summary:**

This paper proposes WaRPAD (Wavelet, Resizing, and Patchifying for AI-generated image Detection), a training-free method for detecting AI-generated images. The core insight leverages the inherent robustness of self-supervised vision models to the RandomResizedCrop  augmentation used during their pre-training.

**Questions:**

See Weaknesses

**Ethical Concerns:**

["NO or VERY MINOR ethics concerns only"]

**Final Justification:**

Borderline accept. The author has solved my concerns.

**Limitations:**

See Weaknesses

**Quality:**

3

**Strengths And Weaknesses:**

Strengths
The paper presents a highly innovative concept – exploiting the inherent RRC invariance learned by self-supervised models on real data as a discriminative signal for AI-generated image detection.

Weaknesses
1. Need Cross-dataset experiments. For example, training on GenImage, and testing on Deepfake-LSUN-Bedroom。
2. While WaRPAD excels on average, Tables 2 & 3 show it can be slightly outperformed by AEROBLADE on specific models (e.g., Midjourney in Synthbuster, SDv1.4 in GenImage). Analyzing the characteristics of images/models where WaRPAD struggles slightly more could be informative.

---

> ### Author Rebuttal · Authors · 2025-07-30
>
> Dear Reviewer g2xN,
>
> We sincerely thank you for your helpful feedback and insightful comments. In what follows, we address your concerns one by one.
>
> ---
>
>
> **[Q1]** Need Cross-dataset experiments. For example, training on GenImage, and testing on Deepfake-LSUN-Bedroom.
>
> **[A1]**
> We would like to clarify that our framework, WaRPAD, is a training-free method that does not require any training images. Therefore, when using our framework, the cross-dataset performance is identical to the reported results in our submission. In contrast, training-based/fine-tuning methods like AIDE [1] and FatFormer [2] could be very sensitive under the cross-dataset experiments. This is the main advantage of training-free methods, including our WaRPAD, over training-based ones.
>
> We here examine our WarPAD against AIDE and FatFormer in the aforementioned benchmarks. To be specific, AIDE is a recent training-based method trained on the ImageNet real data and SDv1.4-generated data. FatFormer is a competitive fine-tuning-based method where additional weights are fine-tuned over the CLIP backbone. FatFormer is trained on 4-class (car, cat, chair, horse) LSUN data and ProGAN-generated data. Therefore, examining AIDE and FatFormer on our examined benchmarks (i.e., Synthbuster/GenImage/Deepfake-LSUN-Bedroom) corresponds to a cross-dataset experiment, except for evaluating AIDE on the GenImage benchmark. We use the pre-trained checkpoint from the official GitHub repository for evaluating the baseline methods.
>
> We show the performance of AIDE, FatFormer, and WaRPAD below. WaRPAD consistently improves over FatFormer on average. Since FatFormer is trained on 4-class LSUN data and ProGAN-generated data, it achieves good performance when evaluated in the LSUN domain and when detecting GAN-generated data. However, its performance is limited in detecting diffusion-based models. While AIDE shows good performance in the GenImage benchmark, where the real image is from ImageNet, it loses robustness on other real domains.
>
> ---
>
> (a) AUROC result on Synthbuster benchmark
>
> | Method     | Firefly | GLIDE | SDXL | SDv2 | SD13 | SD14 | DALL-E3 | DALL-E2 | Midj | Avg  |
> |------------|---------|-------|------|------|------|------|---------|---------|------|------|
> | AIDE       | 0.165   | 0.780 | 0.835| 0.642| 0.946| 0.933| 0.415   | 0.426   | 0.688| 0.648|
> | FatFormer  | 0.586   | 0.718 | 0.707| 0.513| 0.486| 0.500| 0.186   | 0.571   | 0.374| 0.516|
> | WaRPAD     | 0.927   | 0.999 | 0.830| 0.775| 0.959| 0.958| 0.422   | 0.930   | 0.702| 0.834|
> ---
>
> (b) AUROC result in GenImage benchmark.
>
> | Method     | ADM   | BIGGAN | GLIDE | MIDJ | SD14 | SD15 | VQDM | Wukong | Avg  |
> |------------|-------|--------|-------|------|------|------|------|--------|------|
> | AIDE       | 0.921 | 0.920  | 0.987 | 0.959| 1.000| 1.000| 0.965| 1.000  | 0.969|
> | FatFormer  | 0.903 | 0.995  | 0.951 | 0.579| 0.780| 0.776| 0.967| 0.824  | 0.847|
> | WaRPAD     | 0.986 | 0.998  | 0.991 | 0.810| 0.940| 0.936| 0.981| 0.924  | 0.946|
> ---
>
> (c) AUROC result in Deepfake-LSUN-Bedroom benchmark
>
> | Method     | ADM   | DDPM  | DIFF-PjGAN | Diff-StyGAN | IDDPM | LDM   | PNDM  | ProGAN | ProjectedGAN | StyleGAN | Avg   |
> |------------|-------|--------|-------------|-------------|--------|--------|--------|--------|---------------|----------|--------|
> | AIDE       | 0.636 | 0.722  | 0.860       | 0.951       | 0.679  | 0.807  | 0.941  | 0.899  | 0.910         | 0.840    | 0.825  |
> | FatFormer  | 0.745 | 0.709  | 0.998       | 1.000       | 0.824  | 0.944  | 0.999  | 1.000  | 0.999         | 0.988    | 0.921  |
> | WaRPAD     | 0.785 | 0.937  | 0.988       | 0.965       | 0.908  | 0.940  | 0.970  | 0.995  | 0.986         | 0.870    | 0.934  |
> ---
>
>
>
> **[Q2]** While WaRPAD excels on average, Tables 2 & 3 show it can be slightly outperformed by AEROBLADE on specific models (e.g., Midjourney in Synthbuster, SDv1.4 in GenImage). Analyzing the characteristics of images/models where WaRPAD struggles slightly more could be informative.
>
>
> **[A2]**  Thank you for the suggestion. Comparing AEROBLADE and WaRPAD directly can be challenging since they are based on different backbone models: AEROBLADE uses an autoencoder (e.g., VAE), and WaRPAD uses DINOv2. Specifically, AEROBLADE utilizes the reconstruction distance between the input image and its reconstruction via the autoencoder of the latent diffusion model. If the generated image is from the examined autoencoder, it will show a lower reconstruction distance under that autoencoder. Hence, for LDM-based models like SDv1.4/SDv1.5/Wukong, where the generative model’s autoencoder is the same as the examined autoencoder  (we use SDv1.4 for the examination), AEROBLADE ought to show good performance. While Midjourney is not directly analyzable since it is a proprietary model, the results suggest that AEROBLADE may also be specialized in detecting Midjourney-generated data compared to our DINOv2 backbone.
>
> ---
>
> **References**
>
> [1] A Sanity Check for AI-generated Image Detection, Yan et al., ICLR 2025.
>
> [2] Forgery-aware Adaptive Transformer for Generalizable Synthetic Image Detection, Liu et al., CVPR 2024.

---

> > ### Comment · Reviewer_g2xN · 2025-08-07
> >
> > Thank the authors for your answers to my questions. They have clarified most of my concerns and help to confirm my score. In particular, some of your clarifications and additional results in your rebuttal add value to your paper. You should incorporate your answers to my questions in your final version of the manuscript. It will improve your paper and give more insight to potential readers.

---

> > > ### Author Response · Authors · 2025-08-07
> > >
> > > Thank you for your comments and for taking the time to review our manuscript. We are glad that most of your concerns have been clarified.
> > >
> > > We also acknowledge that the reviewer's feedback helped to improve our paper. We will incorporate our additional results and analysis in the final version of the paper.
> > >
> > > Thank you again for your time!

---

### Official Review · Reviewer_Lbu3 · 2025-07-02

**Clarity:** 4
**Significance:** 3
**Originality:** 3
**Rating:** 5
**Confidence:** 5

**Summary:**

This paper presents WaRPAD, a training-free method for detecting AI-generated images using self-supervised models. The key idea is that real images produce more stable embeddings under RandomResizedCrop (RRC) transformations, while synthetic images are more sensitive to such changes—especially in high-frequency areas. WaRPAD uses Haar wavelet decomposition to measure this sensitivity and computes a detection score by analyzing patches of resized images. The method works without any training, performs well across many generative models and datasets, and shows strong robustness to test-time image corruptions.

**Questions:**

The weakness is not really a weakness. But I would like to see some comparisons with supervised detectors on most recent gen-ai model (Flux for example).

**Ethical Concerns:**

["NO or VERY MINOR ethics concerns only"]

**Limitations:**

No negative societal impact.

**Paper Formatting Concerns:**

No concerns.

**Quality:**

3

**Strengths And Weaknesses:**

Strength:
1. The paper presents a clear and well-motivated problem, making it easy to follow the overall idea and reasoning behind the proposed approach.
2. It proposes a training-free detection pipeline that combines RandomResizedCrop-based perturbation and Haar wavelet analysis to compute embedding sensitivity scores on self-supervised models. The method is simple yet effective, and the experimental results are very promising across various generative models.
3. The ablation study is thorough and convincing, demonstrating the contribution of each component and validating the effectiveness of the proposed design.

Weakness:
While the paper compares against several training-free methods, it does not benchmark against recent supervised detectors or fine-tuned models, which would help establish the trade-off between accuracy and efficiency more clearly.

---

> ### Author Rebuttal · Authors · 2025-07-30
>
> Dear Reviewer Lbu3,
>
> We sincerely thank you for your helpful feedback and insightful comments. In what follows, we address your concerns one by one.
>
> ---
>
> **[Q1]** While the paper compares against several training-free methods, it does not benchmark against recent supervised detectors or fine-tuned models, which would help establish the trade-off between accuracy and efficiency more clearly.
>
> **[A1]** We appreciate the suggestion. For the supervised detector, GenImage is a common benchmark, and we compare our WaRPAD against AIDE [1], a recent supervised detector trained in ImageNet images and SDv1.4-generated images. For the fine-tuned model, we compare ours against FatFormer[2], which integrates a training module alongside the CLIP model. FatFormer is trained on the 4-class LSUN (car, cat, chair, horse) images and ProGAN-generated images.
>
> We present the results below. WaRPAD consistently improves over FatFormer on average. While FatFormer relatively performs well on a similar LSUN domain and detecting GAN-generated data, it loses robustness when evaluated under the discrepant real domains or when evaluated on detecting diffusion-generated data. On the other hand, WaRPAD performs comparable to AIDE on the GenImage benchmark with a gap of 2.3% and outperforms AIDE in other benchmarks with a larger gap.
>
>
> ---
>
> (a) AUROC result on Synthbuster benchmark.
>
> | Method     | Firefly | GLIDE | SDXL | SDv2 | SD13 | SD14 | DALL-E3 | DALL-E2 | Midj | Avg  |
> |------------|---------|-------|------|------|------|------|---------|---------|------|------|
> | AIDE       | 0.165   | 0.780 | 0.835| 0.642| 0.946| 0.933| 0.415   | 0.426   | 0.688| 0.648|
> | FatFormer  | 0.586   | 0.718 | 0.707| 0.513| 0.486| 0.500| 0.186   | 0.571   | 0.374| 0.516|
> | WaRPAD     | 0.927   | 0.999 | 0.830| 0.775| 0.959| 0.958| 0.422   | 0.930   | 0.702| 0.834|
> ---
>
> (b) AUROC result in GenImage benchmark.
>
> | Method     | ADM   | BIGGAN | GLIDE | MIDJ | SD14 | SD15 | VQDM | Wukong | Avg  |
> |------------|-------|--------|-------|------|------|------|------|--------|------|
> | AIDE       | 0.921 | 0.920  | 0.987 | 0.959| 1.000| 1.000| 0.965| 1.000  | 0.969|
> | FatFormer  | 0.903 | 0.995  | 0.951 | 0.579| 0.780| 0.776| 0.967| 0.824  | 0.847|
> | WaRPAD     | 0.986 | 0.998  | 0.991 | 0.810| 0.940| 0.936| 0.981| 0.924  | 0.946|
> ---
>
> (c) AUROC result in Deepfake-LSUN-Bedroom benchmark.
>
> | Method     | ADM   | DDPM  | DIFF-PjGAN | Diff-StyGAN | IDDPM | LDM   | PNDM  | ProGAN | ProjectedGAN | StyleGAN | Avg   |
> |------------|-------|--------|-------------|-------------|--------|--------|--------|--------|---------------|----------|--------|
> | AIDE       | 0.636 | 0.722  | 0.860       | 0.951       | 0.679  | 0.807  | 0.941  | 0.899  | 0.910         | 0.840    | 0.825  |
> | FatFormer  | 0.745 | 0.709  | 0.998       | 1.000       | 0.824  | 0.944  | 0.999  | 1.000  | 0.999         | 0.988    | 0.921  |
> | WaRPAD     | 0.785 | 0.937  | 0.988       | 0.965       | 0.908  | 0.940  | 0.970  | 0.995  | 0.986         | 0.870    | 0.934  |
>
> ---
>
> **[Q2]** Comparisons with supervised detectors on most recent gen-ai model (Flux for example).
>
>
> **[A2]** Thank you for the insightful suggestion. To investigate further, we construct a toy dataset using the “Jack-o-lantern” class from ImageNet, chosen because images in this category often feature black backgrounds, consistent with the scenario raised in **[Q3]** of the reviewer 39Dx. Specifically, we generate 1134 synthetic images using the "black-forest-labs/FLUX.1-schnell" model with the prompt “A photo of a jack-o-lantern”, and compare them against 1134 real images from the same ImageNet class.
>
> In terms of AUROC performance, AIDE, WaRPAD, and FatFormer achieve scores of 0.995, 0.962, and 0.354, respectively. WaRPAD shows a relatively comparable performance to AIDE in this specific setting without any fine-tuning.
>
> ---
>
> **References**
>
> [1] A Sanity Check for AI-generated Image Detection, Yan et al., ICLR 2025.
>
> [2] Forgery-aware Adaptive Transformer for Generalizable Synthetic Image Detection, Liu et al., CVPR 2024.

---

> > ### Comment · Reviewer_Lbu3 · 2025-08-06
> >
> > The rebuttal has addressed all my concerns. I will keep the score to accept the paper. Just want to make sure all the contents in the rebuttal will be shown in the final version.

---

> ### Author Response · Authors · 2025-08-06
>
> Thank you for your comments and for taking the time to review our manuscript. We are glad that our rebuttal has addressed your concerns.
>
> We assure that we will include the zero-shot results of the supervised/fine-tuned methods done in the rebuttal, as well as the experiment on the FLUX-generated data, in the main manuscript of the final version of the paper.
>
> Thank you again for your time!

---

### Official Review · Reviewer_39Dx · 2025-07-07

**Clarity:** 3
**Significance:** 3
**Originality:** 2
**Rating:** 4
**Confidence:** 3

**Summary:**

The paper proposes WaRPAD, a training-free method for detecting AI-generated images by leveraging self-supervised models (e.g., DINOv2, CLIP) and their robustness to RandomResizedCrop (RRC) augmentation. It measures sensitivity to high-frequency perturbations via Haar wavelet decomposition. It uses multi-scale patching to simulate RRC effects, achieving strong performance and robustness across diverse generative models and image corruptions.

**Questions:**

See weakness.

**Ethical Concerns:**

["NO or VERY MINOR ethics concerns only"]

**Final Justification:**

The rebuttal has addressed most of my concerns with clarifications or additional experiments. The author should incorporate all the information in the rebuttal into the final version.

**Limitations:**

Yes.

**Paper Formatting Concerns:**

None.

**Quality:**

2

**Strengths And Weaknesses:**

Strength
1) The paper is well-constructed, and the writing is of good quality.
2) The method is simple but effective. The experiment is extensive, and the evaluation results seem good.
3) The method is a training-free method and does not require additional training data. These advantages address a real open problem in the generated image detection domain.

Weakness
1) The main idea of WaRPAD is quite similar to RIGID, which limits its novelty.
2) The motivation of HFwav(x) (Line-148) appears unclear. I would like the authors to provide additional theoretical analysis demonstrating how HFwav(x) reflects the model's robustness to RRC.
3) As shown in Figure 2, the high-frequency components seem to correlate primarily with the proportion of detailed content in images. If this is the case, would the paper's hypothesis only apply to images with abundant details? For instance, when both real and generated images have low content complexity (i.e., fewer high-frequency details), would WaRPAD remain effective?
4) Considering that ImageNet and LSUN (real-world datasets) images undergo post-processing operations, while synthetic data might not receive similar preprocessing, the effectiveness of WaRPAD might stem from detecting post-processing artifacts rather than synthetic artifacts. Notably, WaRPAD focuses on high-frequency differences between real and fake images. However, in the robustness experiments, the authors applied the same post-processing intensity to both real and synthetic images. This could still preserve inherent differences in post-processing characteristics (e.g., noise levels) between datasets.

---

> ### Author Rebuttal · Authors · 2025-07-30
>
> Dear Reviewer 39Dx,
>
> We sincerely thank you for your helpful feedback and insightful comments. In what follows, we address your concerns one by one.
>
> ---
> **[Q1]** The main idea of WaRPAD is quite similar to RIGID, which limits its novelty
>
> **[A1]** We politely disagree with the reviewer’s assessment. Our WaRPAD differs from RIGID in the following perspectives: (1) the introduction of a patch-level robustness evaluation that captures localized embedding sensitivity, and (2) the perturbation strategy for robustness assessment. This is conceptually and methodologically distinct from prior works such as RIGID [1].
>
> First, unlike RIGID, which operates on a global image-level invariance, our WaRPAD analyzes local robustness by evaluating the average embedding stability across spatial patches processed by RescaleNPatchify. This patch-based analysis introduces a novel granularity that has not been explored in prior training-free methods.
>
> Second, our HFWav score quantifies sensitivity across Haar Wavelet-decomposed directions, targeting high-frequency inconsistencies. In contrast, RIGID uses Gaussian noise augmentation.
>
> These distinctions are empirically validated in our ablation study (Table 5), where simply combining RIGID with our patchification (RIGID + RescaleNPatchify) yields minimal improvement. In contrast, our full method, which integrates both the HFwav score and patch-based robustness, achieves significant performance gains, highlighting the novelty and necessity of our design.
>
> ---
> **[Q2]** The motivation of HFwav(x) (Line-148) appears unclear. I would like the authors to provide additional theoretical analysis demonstrating how HFwav(x) reflects the model's robustness to RRC
>
> **[A2]** The HFwav score is designed to approximate a model’s robustness to RRC, particularly focusing on the high-frequency distortions introduced by the resizing component of RRC. Since directly applying RRC introduces stochasticity and multiple hyperparameters, we instead propose HFwav as a deterministic proxy that captures similar effects.
>
> We clarify our motivation in two parts:
> - First, it is a well-established result in image processing that upsampling operations (which are always the case in RRC) distort high-frequency components of an image [2]. Specifically, the upsampling operation cannot faithfully reconstruct the original high-frequency signals [3,4]. These insights are empirically supported by our results in Figure 2, where upsampling distorts the high-frequency component of the original image.
> - Furthermore, DINOv2 is trained to be invariant to RRC augmentations on real images. Therefore, its embedding is expected to be robust to spatial high-frequency perturbations in real images. In contrast, AI-generated images exhibit different high-frequency signals [5,6] and are not part of DINOv2’s training data and thereby may exhibit unstable embeddings under such high-frequency perturbations. HFwav simulates this behaviour by measuring the directional sensitivity to high-frequency components (via wavelet decomposition), allowing us to approximate this differential robustness without explicitly applying RRC.
>
> Aligned with the above motivation, we provide a theoretical analysis of the HFwav function. When the perturbation weight $\alpha$ is small, we can approximate the normalized model output $g_{\text{M}}(\mathbf{x}-\alpha \text{HF}(\mathbf{x}))$ using the first-order Taylor series expansion:
>
> $g_{\text{M}}(\mathbf{x} - \alpha \text{HF}(\mathbf{x})) \approx g_{\text{M}}(\mathbf{x}) - \alpha J_{g_{\text{M}}}(\mathbf{x}) \text{HF}(\mathbf{x})$,
>
> where $g_{\text{M}}(\mathbf{x}) = \text{norm} (f_{\text{M}}(\mathbf{x}))$ is the $\ell_{2}$-normalized model representation and $J_{g_{\text{M}}}(\mathbf{x})$ denotes its Jacobian.
>
> Under this approximation, the HFwav score becomes:
>
> $\text{HFwav}(\mathbf{x}) \approx 1 - \alpha g_{\text{M}}^{\top}(\mathbf{x}) J_{g_{\text{M}}}(\mathbf{x}) \text{HF}(\mathbf{x})$.
>
> where the latter term $ g_{\text{M}}^{\top}(\mathbf{x}) J_{g_{\text{M}}}(\mathbf{x}) \text{HF}(\mathbf{x})$ is the inner product between the embedding $g_{\text{M}}(\mathbf{x})$ and the directional derivative of $g_{\text{M}}$ along the high-frequency direction $\text{HF}(\mathbf{x})$. This term quantifies the embedding shifts in response to its high-frequency perturbation. Following our motivation, the shift is expected to be smaller for real images (due to RRC-invariance) and larger for AI-generated images.
>
> ---
> **[Q3]** As shown in Figure 2, the high-frequency components seem to correlate primarily with the proportion of detailed content in images. If this is the case, would the paper's hypothesis only apply to images with abundant details? For instance, when both real and generated images have low content complexity (i.e., fewer high-frequency details), would WaRPAD remain effective?
>
> **[A3]**  To simulate the scenario in which both real and generated images contain low content complexity (i.e., reduced high-frequency details), we apply an augmentation that center-crops the image by half along both width and height, resulting in a central ¼ area of the original image, and fills the remaining ¾ area with black pixels. This transformation significantly reduces the proportion of detailed content while preserving the structure of the image.
> We evaluate WaRPAD and four baseline methods: RIGID, MINDER, AEROBLADE, and Manifold Bias, on this augmented dataset using the Synthbuster benchmark. The results are presented below, with the comparison against the unaugmented data as the reference.
>
> RIGID and MINDER, which rely on global image perturbation sensitivity, exhibit noticeable performance degradation under reduced content. A similar phenomenon is observed in AEROBLADE, which measures the image reconstruction error of the global image. In contrast, WaRPAD remains relatively robust. This robustness stems from our patch-level examination, where regions with high-frequency details are assessed independently. These results highlight WaRPAD’s resilience to content simplification.
>
> | |RIGID|MINDER|AEROBLADE|Manifold Bias|WaRPAD|
> |-|--|-|-|-|-|
> |Original|0.587|0.518|0.703|0.565|0.834|
> |Fewer details|0.521|0.473|0.663|0.580|0.834|
>
> ---
> **[Q4]** Considering that ImageNet and LSUN images undergo post-processing operations, while synthetic data might not receive similar preprocessing, the effectiveness of WaRPAD might stem from detecting post-processing artifacts rather than synthetic artifacts. Notably, WaRPAD focuses on high-frequency differences between real and fake images. However, in the robustness experiments, the authors applied the same post-processing intensity to both real and synthetic images. This could still preserve inherent differences in post-processing characteristics between datasets
>
> **[A4]** As shown in Figure 8 of the Appendix (available in the supplementary material), we have already included a robustness experiment using the Synthbuster benchmark, where the real data is from the RAISE-1k dataset. This dataset consists of **unprocessed, uncompressed RAW images**. WaRPAD consistently outperforms other training-free methods under this setting across various corruptions on both real and fake data, suggesting that its performance advantage is not primarily due to detecting post-processing artifacts.
>
> To measure the independent effects of post-processing artifacts on real and fake data, we conduct additional experiments using asymmetric corruption scenarios. First, we consider a scenario where real images remain unaltered and AI-generated images are corrupted. Second, we consider a scenario where real images are corrupted and AI-generated images remain unaltered.  All experiments are performed in the Synthbuster benchmark, with JPEG, Crop, and Gaussian noise, with corruption intensity of p denoted as J(p), C(p), and N(p), respectively.
>
> Where only AI-generated images are corrupted, WaRPAD consistently achieves the best performance. While WaRPAD’s performance is reduced by the JPEG compression or Gaussian noise addition of the AI-generated images, a similar phenomenon occurs with other baselines as well. Moreover, RIGID and AEROBLADE exhibit severe performance degradation in the Gaussian noise perturbation.
>
> Where only real images are corrupted, we again observe that AEROBLADE is particularly sensitive to Gaussian noise, indicating that it may be more susceptible to generic image artifacts rather than forensic signals specific to AI-generated content. In contrast, our WaRPAD demonstrates greater robustness across these variations.
>
> ---
>
> (a): AUROC where only the AI-generated data is corrupted
>
> | |0|J(80)|J(60)|C(0.8)|C(0.6)|N(0.05)|N(0.1)|
> |-|-|-|-|-|-|-|-|
> |WaRPAD|0.83|0.71|0.64|0.88|0.84|0.62|0.60|
> |RIGID|0.59|0.54|0.53|0.60|0.63|0.20|0.09|
> |MINDER|0.52|0.51|0.47|0.46|0.47|0.51|0.52|
> |AEROBLADE|0.70|0.61|0.57|0.65|0.69|0.09|0.01|
>
> ---
> (b): AUROC where only the real data is corrupted
>
> | |0|J(80)|J(60)|C(0.8)|C(0.6)|N(0.05)|N(0.1)|
> |-|-|-|-|-|-|-|-|
> |WaRPAD|0.83|0.82|0.85|0.83|0.83|0.87|0.91|
> |RIGID|0.59|0.60|0.62|0.60|0.58|0.64|0.71|
> |MINDER|0.52|0.52|0.51|0.53|0.51|0.51|0.52|
> |AEROBLADE|0.70|0.70|0.68|0.54|0.39|0.99|1.00|
>
> ---
>
> **References**
>
> [1] RIGID, A Training-Free and Model-Agnostic Framework for Robust AI-Generated Image Detection, He et al., ArXiv 2024.
>
> [2] Spatial and Frequency Domain Comparison of Interpolation Techniques in Digital Image Processing, Poth and Szakall,
> International Symposium of Hungarian Researchers on Computational Intelligence and Informatics, 2009.
>
> [3] Improving Feature Stability during Upsampling, Spectral Artifacts and the Importance of Spatial Context, Agnihotri et al., ECCV 2024.
>
> [4] Reversibility Error of Image Interpolation Methods: Definition and Improvements, Briand, IPOL Journal, 2019.
>
> [5] Fourier Spectrum Discrepancies in Deep Network Generated Images, Dzanic et al., NeurIPS 2020.
>
> [6] Leveraging Frequency Analysis for Deep Fake Image Recognition, Frank et al., ICML, 2020.

---

> > ### Comment · Reviewer_39Dx · 2025-08-05
> > **Thanks**
> >
> > The rebuttal has addressed most of my concerns with clarifications or additional experiments. The authors should incorporate all the information in the rebuttal into the final version.

---

> > > ### Author Response · Authors · 2025-08-05
> > >
> > > Thank you for your comments and for taking the time to review our manuscript. We are glad that our additional clarifications and experiments have addressed your concerns. Furthermore, we will incorporate the additional experiments and clarifications in the rebuttal to the final version of the paper.
> > >
> > > If you feel that your concerns have been resolved, we would greatly appreciate it if you could consider updating your rating to reflect your current assessment for the final decision.
> > >
> > > Thank you again for your time!

---

### Official Review · Reviewer_pMdt · 2025-07-08

**Clarity:** 2
**Significance:** 3
**Originality:** 3
**Rating:** 5
**Confidence:** 3

**Summary:**

Paper tackles AI generated image detection. Their contention is that real image distributions that current detection approaches can effectively cover remains extremely limited compared to the diversity of generated images. Growing need for detection methods that can operate universally across diverse domains without relying on the constraints of predefined real image distributions.

The paper leverages the fact that self-supervised models like DINO and DINOv2 are trained (a) solely on real images (b) are adversarially trained to be robust to random cropping and resizing (RRC), via (stronger global view) teacher (weaker augmented) student self-distillation. The main hypothesis of the paper is that embeddings of any AI generated image computed using this self-supervised model would be less robust to random cropping and resizing. This is due to a subsequent hypothesis, which is reasonably well known, that generative models do not model high frequency distributions well (yet).

The paper proposes a Haar-wavelet perturbation sensitivity score – for real images this score is likely to be higher. This is because the self-supervised model is robust to multiple scales and cropping augmentation for real-images. Then for an input image, they rescale it to some random value and patchify it, and average the score over each patch. This score is the WaRPED score acting as a one-class anomaly classifier.

Then the paper compares the methods to other training-free methods and shows improvement. Seems like using a self-supervised model trained to be robust to RRC is key, as their score and rescaling-strategy with RIGID only marginally improves results.

**Questions:**

1.	Why handcraft the objective? Can you show either prior work or comparisons with trained methods that use self-supervised models? It seems like training a fixed DINO with an OOD classifier on top would be even better and achieve zero-shot generalization to all these datasets (this matches recent experience of many recent paper in a variety of domains -- e.g VGGT). The nice thing about a training approach is that it does not rely on a handcrafted signal, hopefully other signals (e.g. biological signals [a]) would not be correctly modelled by generative models when generating images in the future for us to be able to detect them reliably.

2.	Today, generative models do not model high frequency image components well, that may not be true tomorrow. Any suggestions on how to overcome that limitation?

[a] FakeCatcher: Detection of Synthetic Portrait Videos using Biological Signals

**Ethical Concerns:**

["NO or VERY MINOR ethics concerns only"]

**Final Justification:**

Authors have answered most of my concerns, I shall keep my rating.

**Limitations:**

Yes

**Paper Formatting Concerns:**

None.

**Quality:**

3

**Strengths And Weaknesses:**

1.	The motivation and the hypotheses makes sense. DINO and DINOv2 are both trained solely on real data (can be verified from the paper’s data mix), so generated images are OOD. The idea boils down to repurposing DINOv2 weights as a one-class classifier – which is a standard anomaly detection trick. But the challenge is to do this in a training free manner. This line of work makes a lot more sense to me than the ones using diffusion models – self-supervised models are carefully trained on real data (unlike "web scale" diffusion models), and that distribution is likely more constrained. A "membership check" on this distribution is a good way to identify anomalies.

2.	The method also makes intuitive sense. DINO objective is to do self-distillation with an augmented student which is fed local crops and a teacher is which is fed global images – thus it will be robust to cropping and resizing. For an OOD sample, that the invariant is likely to not hold.

3.	Motivation of Haar-wavelet perturbation score. The authors also identify the reason for this OOD behaviour, generative models trained to match real image distributions can’t model high frequency well. This is quite well known. I think they should cite some analysis papers here to support their argument. But it all makes sense.

4.	Comparison and results appear to be fair, they have tested a variety of methods and datasets. Ablations are also reasonable and cover most common questions.

Weaknesses

5.	Exploits a known property of existing generative models, and handcrafts a score function to be training-free. However, I’m not sure if that’s a good idea, why should this OOD classifier be training free and handcrafted, if we have deepfake datasets available to train on. Of course, one can leverage the DINO representations in that case too, by keeping it fixed. However, I’ll not penalize this specific paper too much on this, given that there is prior literature in the “training free” space, and this limitation likely applies to all of them. But I’d like the authors to answer the question for me, as I think this is an important baseline. In their introduction, the contention that “real image distributions that current detection approaches can effectively cover remains extremely limited compared to the diversity of generated images” is not shown.

6.	The ablations and analysis can be more clearly written – the main points do not come out as effectively as the introduction and the method section.

---

> ### Author Rebuttal · Authors · 2025-07-30
>
> Dear Reviewer pMdt,
>
> We sincerely thank you for your helpful feedback and insightful comments. In what follows, we address your concerns one by one.
>
> ---
> **[Q1]** Exploits a known property of existing generative models, and handcrafts a score function to be training-free. However, I’m not sure if that’s a good idea, why should this OOD classifier be training free and handcrafted, if we have deepfake datasets available to train on.
>
> **[A1]** We agree that when abundant real and generated images are available in a given domain, fine-tuning a classifier over the fixed foundation model may achieve stronger in-distribution performance. Indeed, most AI-generated image detection methods train and test on the same (or similar) real image distribution, assuming an abundance of real data. However, the availability of deepfake datasets may not always be the case in AI-generated image detection in practical scenarios, where it is practically impossible to collect data from all real domains.
>
> First, generalization to unseen real domains remains a significant challenge to the training-based/fine-tuning approach. While training-based detectors perform well on datasets they are exposed to (e.g., ImageNet, LSUN, Raise-1k), their effectiveness may deteriorate when deployed in out-of-distribution (OOD) domains such as artwork, medical imaging, or industrial anomaly data. For example, [1] proposes a FakeART benchmark of training on COCO/LSUN real images and examining on WikiART real images, where existing training-based methods (including FatFormer[2] that fine-tunes from CLIP) show underwhelming accuracy smaller than 60%.  Compared to the generalization capability of the DINOv2 backbone weight (which is often regarded as a robust, off-shell detector that show effectiveness on such domains without any fine-tuning [3,4]), the fine-tuned classifier may be more prone to the distribution shift and may require additional fine-tuning on that domain (see **[A3]** for the comparison against FatFormer).
>
> Furthermore, the real image data is often scarce or proprietary, especially in domains like healthcare or industrial anomaly detection. In such domains, relying on a pre-trained proprietary model or a generalist foundation model (like DINOv2) can be the best option for practical purposes. Many training-free approaches, including WaRPAD,  are designed to operate with minimal assumptions about internal model information. For example, WaRPAD only requires access to the output embeddings of a pre-trained, RRC-invariant model. This makes our approach suitable for restricted or black-box environments where model weights are inaccessible.
>
> In summary, training-free approaches leverage versatile backbone models that often exhibit stronger robustness to OOD domains compared to fine-tuned classifiers, particularly in scenarios where training data are scarce or unavailable. Our approach aligns with the core philosophy of robustness-critical frameworks such as domain generalization. We will revise the contention statement in the introduction to more clearly reflect this perspective.
>
> ---
> **[Q2]** The ablations and analysis can be more clearly written - the main points do not come out as effectively as the introduction and the method section.
>
> **[A2]** We thank the reviewer for the helpful suggestion. In the revised manuscript, we will reorganize this section to more clearly highlight the purpose and key findings of each experiment.
>
> ---
>
> **[Q3]** Can you show either prior work or comparisons with trained methods that use self-supervised models?
>
> **[A3]** Thank you for your suggestion. Indeed, several works [2,5] fine-tune the lightweight detection module over the frozen self-supervised model backbone. For example, FatFormer [2] integrates a forgery-aware adapter module along with the image encoder of the CLIP model and a text-guided interactor module along with the text encoder of the CLIP model. We test the author’s official FatFormer checkpoint, which is trained on 4-class (car, cat, chair, horse) LSUN and ProGAN-generated data.
>
> We compare the performance of FatFormer and WaRPAD, where the results are shown below. Overall, FatFormer underperforms over WaRPAD on average. In the LSUN-Bedroom benchmark, FatFormer shows near-perfect performance in detecting GAN-generated data, but the performance is degraded in detecting the diffusion-model-generated data. When the real data is more discrepant from the 4-class LSUN data, the performance gap is widened, and FatFormer consistently underperforms over WaRPAD in all generated data.
>
> Furthermore, we compare FatFormer and WaRPAD in the art-domain data. Since the FakeART benchmark in [1] is not public except for the information that they use the WikiArt dataset for the real data, we instead download data from the Kaggle webpage with 10821/10821 real and AI-generated data available, where the real data is from the WikiArt dataset. In the dataset, FatFormer’s AUROC is 0.531 while WaRPAD shows an AUROC of 0.765.
>
> In summary, the above experiments show that the fine-tuning-based method can underperform in generalizing to unseen real data distribution discrepant from the training real data distribution, which aligns with **[A1]**. While the generalization of such fine-tuning-based models can be improved by integrating more domain data and generative models in the fine-tuning phase, the availability of the training data might be limited in some domains, where our training-free setting truly shines.
>
> ---
>
> (a) AUROC in Synthbuster.
>
> |Method|Firefly|GLIDE|SDXL|SDv2|SD13|SD14|DALL-E3|DALL-E2|Midj|Avg|
> |--|--|--|--|--|--|--|--|--|--|--|
> |FatFormer|0.586|0.718|0.707|0.513|0.486|0.500|0.186|0.571|0.374|0.516|
> |WaRPAD|0.927|0.999|0.830|0.775|0.959|0.958|0.422|0.930|0.702|0.834|
> ---
>
> (b) AUROC in GenImage.
>
> |Method|ADM|BIGGAN|GLIDE|MIDJ|SD14|SD15|VQDM|Wukong|Avg|
> |--|--|--|--|--|--|--|--|--|--|
> |FatFormer|0.903|0.995|0.951|0.579|0.780|0.776|0.967|0.824| 0.847|
> |WaRPAD|0.986|0.998|0.991|0.810|0.940|0.936|0.981|0.924| 0.946|
> ---
>
> (c) AUROC in Deepfake-LSUN-Bedroom.
>
> |Method|ADM|DDPM|DIFF-PjGAN|Diff-StyGAN|IDDPM|LDM|PNDM|ProGAN|ProjectedGAN|StyleGAN|Avg|
> |--|--|--|--|--|--|--|--|--|--|--|--|
> |FatFormer|0.745|0.709|0.998|1.000|0.824|0.944|0.999|1.000|0.999|0.988|0.921|
> |WaRPAD|0.785|0.937|0.988|0.965|0.908|0.940|0.970|0.995|0.986|0.870|0.934|
>
> ---
> **[Q4]** Today, generative models do not model high-frequency image components well, that may not be tomorrow. Any suggestions on how to overcome that limitation?
>
> **[A4]**  We appreciate the reviewer’s perspective. Indeed, when future generative models can succeed in faithfully generating realistic images (including high-frequency components), any detection approach (including ours) may become less effective. We here suggest several research directions to mitigate this limitation and enhance the long-term viability of our approach.
>
> First, our score function can be extended beyond high-frequency wavelet statistics. For example, RRC might affect other spatial frequency bands, not just the high-frequency components. Integrating sensitivity on those mid-frequency bands might improve WaRPAD further. In addition, geometric augmentations (e.g., rotation) that are widely used in OOD detection [6] can be effective in revealing further discrepancies in generated data.
>
> Furthermore, future work may benefit by incorporating multimodal semantic understanding. While we currently operate in a training-free, vision-only setting, emerging multimodal foundation models [7,8] may eventually support zero-shot reasoning about the consistency between an image and its plausible semantics (e.g., physics, anatomy). As seen in some training-based detection methods [1,9], incorporating real-world priors can improve AI-generated image detection performance. We anticipate that multimodal foundation models pre-trained on world-knowledge-aligned data [10,11] could enable such reasoning in a training-free fashion in the future.
>
> In summary, we expect that integrating mid-frequency stability and augmentation-specific inconsistencies at a vision level, and integrating multimodal priors, might be promising extensions to maintain the robustness of our approach. We leave this direction to future work.
>
> ---
> **[Q5]**  The authors identify the reason for this OOD behaviour, generative models trained to match real image distributions can’t model high-frequency well. This is quite well known. I think they should cite some analysis papers here to support their argument.
>
> **[A5]** Thank you for the suggestions. We will add citations of the related papers [12,13] in the revised manuscript.
>
>  ---
> **References**
>
> [1] Secret Lies in Color: Enhancing AI-Generated Images Detection with Color Distribution Analysis, Jia et al., CVPR 2025.
>
> [2] Forgery-aware Adaptive Transformer for Generalizable Synthetic Image Detection, Liu et al., CVPR 2024.
>
> [3] DINOSim: Zero-Shot Object Detection and Semantic Segmentation on Electron Microscopy Images, BioRxiv 2025.
>
> [4] AnomalyDINO: Boosting Patch-based Few-shot Anomaly Detection with DINOv2, Damm et al., WACV 2025.
>
> [5] Towards Universal Fake Image Detectors that Generalize Across Generative Models, Ohja et al., CVPR 2023.
>
> [6] Using Self-Supervised Learning Can Improve Model Robustness and Uncertainty, Hendrycks et al., NeurIPS 2019.
>
> [7] Magma: A Foundation Model for Multimodal AI Agents, Yang et al., CVPR 2025.
>
> [8] BAGEL: The Open-Source Unified Multimodal Model, Deng et al., ArXiv 2025.
>
> [9] FakeCatcher: Detection of Synthetic Portrait Videos using Biological Signals, TPAMI 2020.
>
> [10] WISE: A World Knowledge-Informed Semantic Evaluation for Text-to-Image Generation, Niu et al., ArXiv 2025.
>
> [11] Envisioning Beyond the Pixels: Benchmarking Reasoning-Informed Visual Editing, Zhao et al., ArXiv 2025.
>
> [12] On the detection of synthetic images generated by diffusion models, Corvi et al., ICASSP 2023.
>
> [13] On the Frequency Bias of Generative Models, Schwarz et al., NeurIPS 2021.

---

> > ### Comment · Reviewer_pMdt · 2025-07-31
> > **Thanks for the response**
> >
> > Answers my questions, thanks! I see that the fine tuned Fatformer numbers are zero-shot, do mention that. Did you try any experiments with images generated by the latest FLUX model?

---

> ### Author Response · Authors · 2025-08-01
>
> Thank you for the quick response.
>
> Indeed, our examination of the FatFormer in this rebuttal is conducted in the zero-shot setting, and we will explicitly mention this.
>
> For the comparison against FLUX, we further test WaRPAD and FatFomer in zero-shot when the real image is from the "Jack-o'-lantern" class on the ImageNet dataset, and AI-generated images are from the "black-forest-labs/FLUX.1-schnell" model under the prompt "A photo of a jack-o'-lantern".  WaRPAD scores 0.962 AUROC while FatFormer scores 0.354. We also refer to our response to Reviewer Lbu3 for further information.

---

> > ### Comment · Reviewer_pMdt · 2025-08-01
> > **Thanks!**
> >
> > Thanks, this helps my judgement. :)

---

### Author Response · Authors · 2025-08-05

Dear AC and Reviewers,

We hope this message finds you well. We are writing to kindly follow up regarding our rebuttal.

We have made a sincere effort to address the reviewers' concerns through detailed clarifications, additional experiments, and further analysis. We believe these additions meaningfully strengthen our paper and help clarify important points.

If there are any remaining concerns or unresolved issues, please feel free to let us know. We would be happy to provide further clarification.

Thank you again for your time and valuable feedback.

Best regards, \
Authors

---

### Decision · Program_Chairs · 2025-09-17

**Decision:**

Accept (poster)

**Comment:**

During discussion period, the authors and reviewers have comprehensive discussions, and all reviewers agree to accept this paper.